# Enhancements of electric field and afterglow of non-equilibrium plasma by Pb(Zr$_x$Ti$_{1-x}$)O$_3$ ferroelectric electrode

Yijie Xu [1], Ning Liu [1] ✉, Ying Lin[1], Xingqian Mao[1], Hongtao Zhong[1,3], Ziqiao Chang [1], Mikhail N. Shneider [1] & Yiguang Ju [1,2]

Manipulating surface charge, electric field, and plasma afterglow in a non-equilibrium plasma is critical to control plasma-surface interaction for plasma catalysis and manufacturing. Here, we show enhancements of surface charge, electric field during breakdown, and afterglow by ferroelectric barrier discharge. The results show that the ferroelectrics manifest spontaneous electric polarization to increase the surface charge by two orders of magnitude compared to discharge with an alumina barrier. Time-resolved in-situ electric field measurements reveal that the fast polarization of ferroelectrics enhances the electric field during the breakdown in streamer discharge and doubles the electric field compared to the dielectric barrier discharge. Moreover, due to the existence of surface charge, the ferroelectric electrode extends the afterglow time and makes discharge sustained longer when alternating the external electric field polarity. The present results show that ferroelectric barrier discharge offers a promising technique to tune plasma properties for efficient plasma catalysis and electrified manufacturing.

Non-equilibrium low-temperature plasma, powered by renewable electricity, is a promising technology for distributed and green manufacturing, including electrified catalysis[1,2], material synthesis[3,4], fuel reforming[5,6], waste materials recycling and upcycling[7] and pollution reduction[8,9], etc. Non-equilibrium plasma can generate energetic electrons, ions, excited species[10], and surface charges to initiate and facilitate chemical reactions either in the gas phase or on gas/solid/liquid surfaces with reduced kinetic barriers and different non-equilibrium pathways at low temperatures[11,12]. Furthermore, low-temperature plasma is an ideal technology to address the intermittency challenge of renewable energy produced from wind and solar[13] by enabling fast and durable energy storage in synthesizing e-fuels such as hydrogen[5], ammonia[14], alcohol[15] and materials like cement[16] and steel[17]. Taking the plasma-catalytic ammonia synthesis as an example, low-temperature plasma enhances ammonia production at lower pressures and temperatures than the traditional Haber-Bosch process by creating vibrationally and electronically excited species as well as active radicals via electron impact energy transfer and enabling different reaction pathways and catalyst design [14].

Despite the above advantages, low-temperature plasma manufacturing still faces some challenges. Firstly, there is a lack of control methods of electric field and near-surface active species (e.g., electrons, ions, excited and polarized species) to tune plasma chemistry and significantly enhance the reactivity near the interface between plasma and solids/liquids[18]. For instance, due to the low electric field and reactive species concentrations near the surface, plasma-catalytic ammonia synthesis is at least one order of magnitude less energy efficient than conventional processes (25–30 g-NH$_3$ kWh$^{-1}$ [19] compared to 500 g-NH$_3$ kWh$^{-1}$ for large-scale Haber−Bosch process[20]). Secondly, there is a need to utilize and control afterglow to reduce the quenching of transient reactive species and further enhance energy conversion efficiency. Currently, most industrial plasma power supplies, e.g., microwave, radio frequency (RF) or alternating current (AC), contain a great amount of inter-cycle time with very short-lived afterglow (i.e., in

[1]Department of Mechanical and Aerospace Engineering, Princeton University, Princeton, NJ 08540, USA. [2]Princeton Plasma Physics Laboratory, Princeton, NJ 08540, USA. [3]Present address: Department of Mechanical Engineering, Stanford University, Stanford, CA 94305, USA. ✉e-mail: nl7@princeton.edu

low/zero external electric field ranges). During afterglow, many reactive species (e.g., electrons, ions and excited species) are rapidly quenched within hundreds of nanoseconds (ns), disabling plasma reactivity and lowering its efficiency. Therefore, reviving and controlling afterglow will be of great potential to enhance plasma reactivity through sustaining transient reactive species.

To address these challenges, incorporating ferroelectric materials into plasma manufacturing will be promising. Ferroelectric materials with asymmetric crystal structures inherently feature spontaneous electric polarization under an external electric field in ferroelectric barrier discharge (FBD). And such ferroelectric polarization is highly tunable towards various plasma manufacturing by manipulating temperature[21] and defects[22], etc. Unfortunately, only a few studies have been conducted to understand the effect of ferroelectric materials on plasma properties and dynamics. For example, electrical properties, including voltages and currents[23] and discharge patterns of both gas[24] and surface[25,26] were preliminarily characterized in FBD plasmas. Other studies mainly focused on practically applying ferroelectric materials to improve plasma manufacturing yield (e.g., ammonia synthesis[27,28], $NO_x$ removal[29], and carbon dioxide conversion[30]). Therefore, the underlying physics of how ferroelectric materials enhance plasma properties and discharge is still unclear, especially how ferroelectrics-induced surface charge augments electric field and instabilities in plasmas, and whether surface charge further holds discharge on in afterglow plasma. These open questions have been bottlenecking the development of FBD plasma manufacturing till now.

As such, this work aims to advance the fundamental understanding of FBD on plasma properties such as surface charge, electric field, and afterglow. We first report the significant enhancement of surface charge by ferroelectric materials. Then, with time-resolved in-situ electric field induced second harmonic generation and Sawyer-Tower measurements, we discover the underlying physics of the electric field overshoot during the FBD breakdown via fast ferroelectric polarization compared to streamer breakdown. Moreover, we discover that compared to the conventional dielectric barrier discharge (DBD), the ferroelectrics-induced surface charge could extend afterglow time by 68 times and make FBD plasma (e.g., AC plasma as demonstrated in this work) much more temporally sustainable when alternating the external electric field polarity. Overall, the high electric field during breakdown and extended afterglow of FBD enable stronger streamers and reduce quenching of transient reactive species.

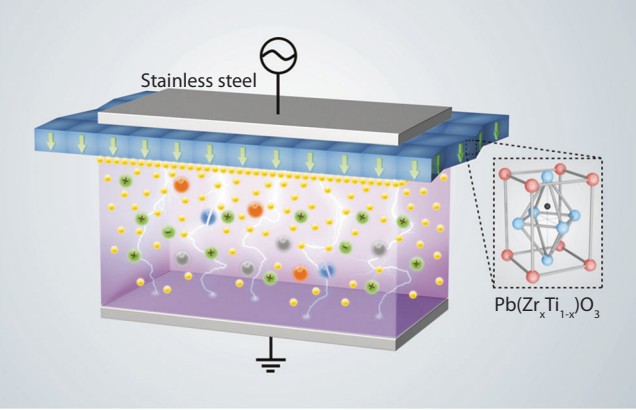

**Fig. 1 | Ferroelectric barrier discharge (FBD) design.** Two plate electrodes are placed in parallel: one is assembled by attaching a ferroelectric Pb($Zr_xTi_{1-x}$)O₃ (PZT) layer to the stainless steel and the other one is a pure stainless steel plate. The asymmetric tetragonal crystal structure of PZT results in the spontaneous polarization and accumulation of charges near the surface of the PZT. Because of the formation of a large surface charge, the plasma is significantly strengthened with an enlarged electric field and increasing densities of charged particles and excited species in the plasma.

These unique features are particularly promising for high-efficiency plasma manufacturing (e.g., catalysis and synthesis).

## Results

### The design of the ferroelectric barrier discharge (FBD)

As shown in Fig. 1, the plasma was generated between two parallel plate electrodes. One electrode was assembled by attaching a ferroelectric material layer (45 mm × 17 mm × 1.2 mm) to a stainless-steel plate, and the other was a pure stainless-steel plate. These two electrodes were separated by a 5 mm distance and were placed into a rectangular quartz cell. The ferroelectric layer uses lead zirconate titanate (Pb($Zr_xTi_{1-x}$)O₃, PZT), which is a ceramic perovskite material with a tetragonal crystal structure (under $T_c \approx 320$ °C), and the asymmetric locations of the cations and anions result in the separation of positive and negative charge centers inside the unit cell and equip PZT with spontaneous polarization nature[25]. The high purity of the PZT material in this study is confirmed by the X-ray diffraction pattern in Supplementary Fig. 1, and the PZT material was pre-poled to achieve satisfactory ferroelectricity in FBD. The PZT is used here to create fast electric polarization and large surface charge to control plasma properties. The ferroelectric electrode was connected to an AC power supply running at a frequency of 20 kHz and a peak-to-peak voltage of 3 kV, while the pure stainless-steel electrode was grounded. $N_2$ gas flows between electrodes with a residence time of 0.05 s at low pressure ranging between 10 and 100 torr. More information about the FBD design can be found in the Method section.

### Ferroelectric electrode induces large surface charge

We used the Sawyer-Tower method[31] to in-situ measure the surface charge generated by FBD. The Sawyer-Tower method measures the surface charge based on the conservation of charge in a series circuit, and its implementation involves placing a reference capacitor with a known capacitance of 10 nF in series with FBD. The reference capacitance is much larger than the capacitance of the ferroelectric electrode. As a result, the voltage drop on the reference capacitor is negligible compared to that on FBD, and the disturbance caused by the reference capacitor is negligible to the original FBD operation. Figure 2 summarizes the surface charges (denoted as $Q$) of FBD and DBD at different applied voltages ($U_{cell}$). For comparison purposes, we also measured the surface charge in a representative classic dielectric barrier discharge (DBD). This DBD involves replacing the PZT layer with an alumina ($Al_2O_3$) layer of the exact dimensions. More details are stated in the Method section.

As seen in Fig. 2, a significantly large amount of surface charge has been introduced by the ferroelectric PZT barrier and accumulates on the PZT surface compared to the dielectric alumina barrier under different pressure conditions. As shown in Fig. 2a, the surface charge is induced by the ferroelectric PZT at 10 torr, and it achieves 0.1 μC cm⁻² while the DBD depicts a two-orders-of-magnitude lower charge level of 10⁻³ μC cm⁻². With the pressure elevated to 40 torr and 70 torr as in Fig. 2b and Fig. 2c, the peak surface charges induced by ferroelectrics slightly increase to 0.12-0.13 μC cm⁻². We attribute this mainly to an enhanced secondary electron emission (electrons emitted after energetic particles reach the surface of ferroelectrics) of the ferroelectric electrode with increasing concentrations of energetic species (e.g., electrons and ions) under higher pressure[24]. In return, more electrons with higher electron temperatures from the enhanced secondary electron emission in FBD collide with neutral molecules, leading to more energetic excited species (electronic and/or vibrational) and more ions, compared to DBD. Also, the difference in work function contributed to the difference in induced electron concentration between FBD and DBD. PZT has a work function of ~4.5 eV[32], smaller than that (i.e., ~4.7 eV[33]) of alumina. Therefore, the surface of the PZT barrier electrode in FBD directly emits more electrons into the plasma than that of the alumina barrier electrode in DBD, which further

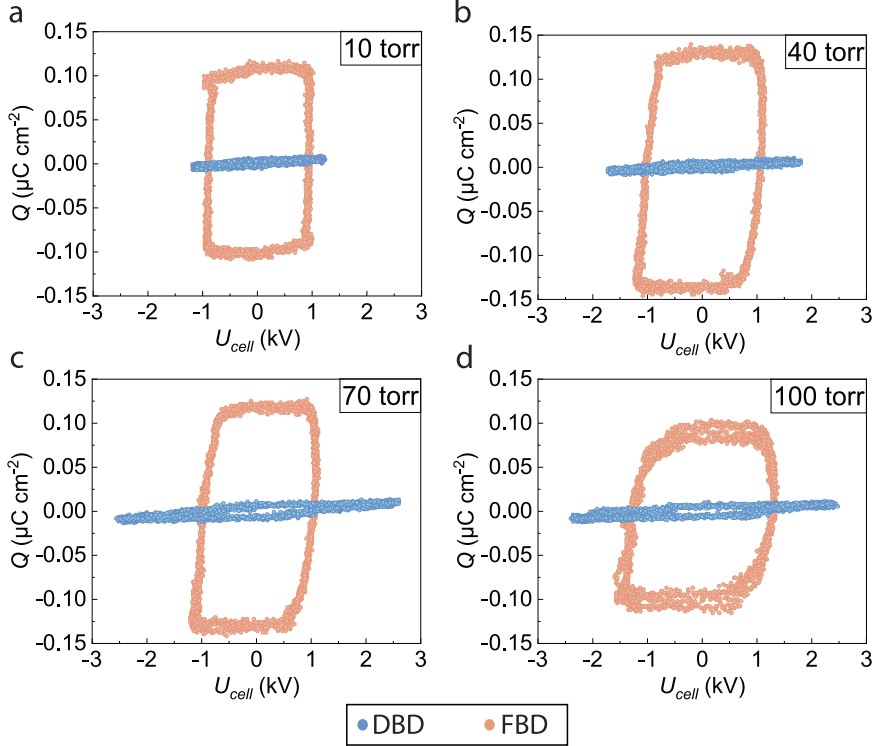

**Fig. 2 | Surface charge measurements at different pressures in ferroelectric barrier discharge (FBD) and dielectric barrier discharge (DBD).** The maximum surface charges ($Q$) of FBD achieves $10^{-1}$ μC cm$^{-2}$, which is two orders of magnitude higher than that of DBD under **a** 10 torr, **b** 40 torr, **c** 70 torr and **d** 100 torr. The $Q(U_{cell})$ plot is directly measured by placing a reference capacitor (10 nF) in series with the plasma reactor.

induces stronger secondary electron emission and eventually leads to a higher electron concentration in FBD than in DBD.

However, the peak surface charge drops back to ~0.1 μC cm$^{-2}$ when the pressure reaches 100 torr as in Fig. 2d. Because the ions and electrons causing secondary emission on the ferroelectric electrode become less energetic with lower mean energy gained from the external electric field in plasma. Also, the concentrations of ions and electrons may drop back, since a higher pressure leads to a lower reduced electric field (*E/N*, defined as the electric field divided by the total number density of molecules) and hence a lower ionization rate, despite a higher number density of molecules. Furthermore, it is noteworthy that the voltage at the electric coercivity of FBD increases as the pressure increases. This is because the voltage at the electric coercivity is approximately equal to the breakdown voltage denoted by the voltage at the point where the magnitude of negative surface charge with the positive applied voltage starts to decrease (i.e., the lower right corner of the curve) in Fig. 2. Based on Paschen's law, the gas breakdown voltage (i.e., the voltage at electric coercivity) increases as the pressure rises.

**Enhanced electric field during breakdown**

The above PZT-induced surface charge significantly enhances plasma properties, as summarized in Fig. 3. Figure 3a shows the voltage waveforms applied on FBD and DBD, while Fig. 3b shows the current waveforms correspondingly. As seen, in each half cycle, the voltage waveform of FBD experiences a drop where simultaneously its current increases sharply. For example, at 93.5 μs, a large voltage drop of 1.65 kV for FBD occurs while the current dramatically increases to ~0.9 A, as shown in the zoom-in view inside Fig. 3b. This indicates a ferroelectric switching during which the accumulated charge is rapidly realigned on the ferroelectric electrode surface[23,24]. Ferroelectric switching, which is an inherent feature of ferroelectric materials, is the process of switching polarized domains by applying an electric field[34].

By contrast, the DBD voltage waveform has negligible distortions, and the current at 93.5 μs is only ~0.14 A. After the first and large voltage drop at 93.5 μs, the FBD voltage waveform experiences small drops with current filaments coexisting when ferroelectric polarization is reversed. For example, the voltage drops at -142 μs and 192 μs are only ~0.3 kV and ~0.1 kV, respectively. This is probably because the voltage applied after 93.5 μs is apparently smaller (balanced by a larger FBD induced current at a fixed power), leading to a weaker ferroelectric switching. Supplementary Fig. 2 shows the zoom-in view of the switching current profile after the first streamer breakdown of FBD. As seen, the switching current profile during the following streamer breakdown is on the same order (i.e., 0.005 A) as the breakdown current of DBD streamers. However, different from DBD which only has one or two current peaks during the breakdown process, FBD has multiple (i.e., 4–7) current peaks, indicating more streamers formed due to larger ferroelectrics-induced surface charge. The switching current after the first switching is much smaller than the first switching current. This is because after the first breakdown, FBD has produced enough charges (in terms of electrons) to form and sustain streamers, and the charges generated from PZT switching are transferred through streamers rather than being detected as a large current peak.

To understand the effect of surface charge induced by rapid polarization of PZT on the electric field and breakdown, we performed time-resolved electric field measurements using Electric Field Induced Second Harmonic (EFISH) generation. In EFISH, the electric field is measured through the second harmonic signal of the excitation laser that quadratically scales with the externally applied electric field strength[35], as detailed in the Method section. Figure 3c shows the electric field obtained along the central line of N$_2$ plasma (i.e., outside the sheath layer) at 100 torr as an example. As seen, initially (before breakdown), the electric field for the FBD is approximately the same as that for the DBD. Then, at the first time of gas breakdown (i.e., 93.5 μs), the electric field of FBD plasma rapidly increases to -13 kV cm$^{-1}$, while

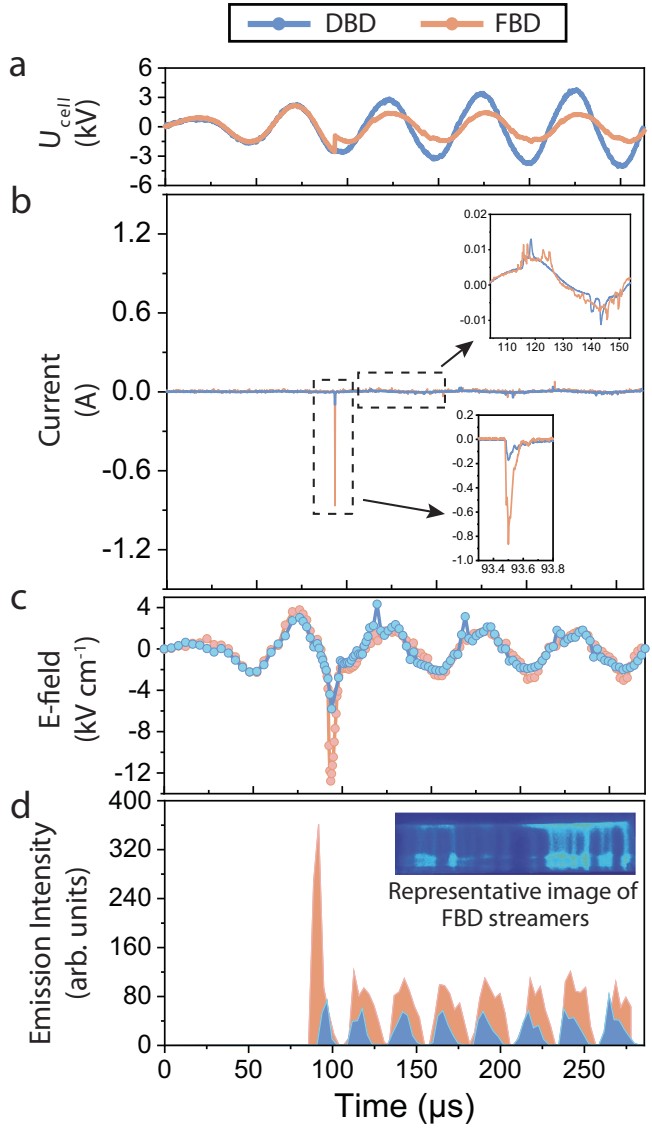

**Fig. 3 | Surface charge enhanced plasma comparing ferroelectric barrier discharge (FBD) to dielectric barrier discharge (DBD). a** Applied voltage $U_{cell}$ waveforms on the reactor in FBD and DBD at 100 torr. The voltage drop indicates a ferroelectric switching during which the accumulated charge is rapidly realigned on the ferroelectric electrode surface causing the gas breakdown. **b** The induced current waveforms on FBD and DBD. A sharp increase can be seen simultaneously where the voltage drops during breakdown. **c** The electric field measurements on FBD and DBD. The electric field is obtained by EFISH diagnostic along the central line of plasma corresponding to the current measurement. The large surface charge in FBD enlarges the electric field during breakdown and sustains the electric field afterward. While the DBD has to experience multiple streamer formations after the first one, indicated by the following spikes when the AC electric field polarity changes. **d** The time history of FBD and DBD emission intensities. The emission intensity further proves that the FBD electrode surface charge can better sustain the plasma when the electric field polarity changes. The hysteresis nature of the surface charge in FBD sustains the streamers, but the DBD with a low surface charge still needs to establish new streamers for new gas breakdown when alternating electric field polarity.

the electric field of DBD only climbs to ~5 kV cm⁻¹. Such large transient electric field differences can be explained by comparing the timescales of ferroelectric polarization against electric breakdown. As suggested in the reference work[23], the timescale of ferroelectric polarization can be characterized by the fall (or rise) time of the transient voltage change process, which was measured to be below 10 ns as shown in

Supplementary Fig. 3. In parallel, with the AC power and dielectric/ferroelectric layer in this work, the plasma is streamer discharge, and its breakdown timescale consists of streamer formation time and propagation time. The former is on the order of several ns[36] and the latter is several tens of ns (given the streamer propagation velocity is $10^5$ m s⁻¹ or less[37] and the distance of the gap is 5 mm). Therefore, the breakdown happens much slower than the surface charge formation dictated by fast ferroelectric polarization, and the formed surface charge instantaneously overshoots the electric field before the breakdown occurs. By contrast, the DBD electric field overshoot, though also exists, is much smaller because the surface charging of DBD (evidenced in Fig. 2d) is much smaller than that of FBD.

Then, as shown in Fig. 3c, when the external electric field polarity changes, the electric field for the FBD plasma does not experience any significant overshoot during the breakdown; by contrast, the electric field for the DBD plasma contains gradually decaying overshoots. We attribute this difference to the large number of seed electrons generated in the first round of discharge and the extended charge lifetime in FBD. Here, the seed electrons are defined as the initial free electrons before gas breakdown and plasma formation, which are the remaining electrons generated from the last plasma discharge after partially quenching. During the first discharge at 93.5 μs, the FBD plasma due to surface charge features a much larger current than the DBD plasma as shown in Fig. 3b, and the emission intensity of the FBD plasma is also much higher than that of DBD as shown in Fig. 3d. As a result, the FBD plasma provides more seed electrons to further accelerate its following streamer formation and propagation to the timescale of ferroelectric switching (i.e., ~10 ns), which avoids electric field overshoot as mentioned above. By contrast, the DBD still needs breakdown (shown in Fig. 3c) to form streamer discharges and accumulate seed electrons. Before accumulating sufficient seed electrons, the overshoot of the DBD electric field during breakdown still exists but gradually decays.

Furthermore, we explored the influence of dielectric constant on the electric field. This was achieved by performing EFISH measurements on the electrode covered by $SrTiO_3$ (STO), a paraelectric material with a dielectric constant of 300 between PZT and alumina. For the EFISH measurements on the $SrTiO_3$ barrier electrode, all the experimental conditions (e.g., flow rate and 100 torr pressure) or parameters (e.g., dimensions) remain the same as those for either PZT FBD or alumina DBD, except the electrode barrier material itself. The results are shown in Supplementary Fig. 4a. As shown, the electric field for the STO barrier discharge lies between the electric fields of PZT FBD and alumina DBD, because the STO's dielectric constant of 300 between that of PZT and alumina, and the surface charge induced by STO is also between those by PZT and alumina. More specifically, with an intermediate surface charge level, the electric field overshoot of the STO barrier discharge during the first gas breakdown reaches ~9 kV cm⁻¹, which is smaller than that of PZT FBD but larger than alumina DBD. Correspondingly, the concentration of the remaining electrons offered by the first streamer discharge and serving as seed electrons lies between those of PZT FBD and alumina DBD. As a result, the electric field overshoot of the STO barrier discharge during the following breakdown cycles also lies between those of PZT FBD and alumina DBD, i.e., decaying faster than the alumina DBD electric field but slower than the PZT FBD electric field.

Two notes are worth mentioning here on EFISH measurements. First, the above EFISH measurements were performed at 100 torr as an example to demonstrate the electric field enhancement in FBD. Due to its electric field detection limit (e.g., 0.41 kV cm⁻¹ at 100 torr), the EFISH measurements were not practically feasible at low pressures (e.g., 10 torr). Second, the temperature change on the PZT barrier electrode surface during gas breakdown and plasma formation is negligible and is within the measurement uncertainty (~0.1 °C) based on thermal imaging to be mentioned later. The surface charge of PZT caused by pyroelectricity[38] during gas breakdown and plasma formation is

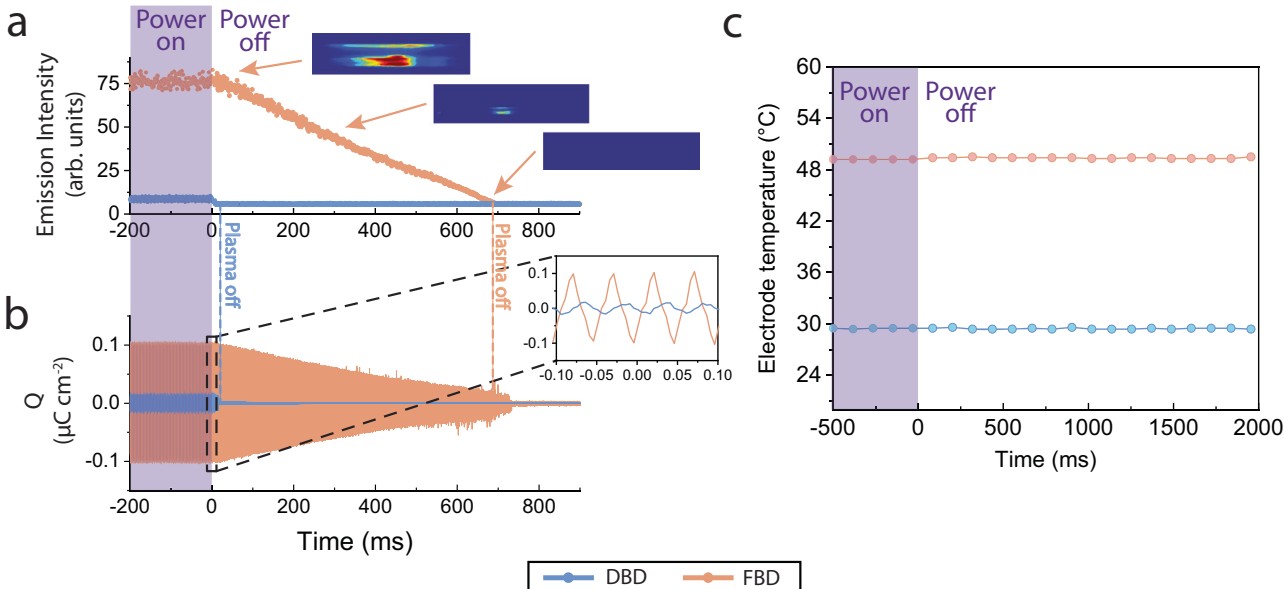

**Fig. 4 | Sustained ferroelectric barrier discharge (FBD) plasma afterglow. a** The time histories of FBD and dielectric barrier discharge (DBD) emission intensity afterglow. The emission intensity of FBD decays gradually with a shrinking plasma volume and lasts for ~680 ms. By contrast, the DBD plasma quickly extinguishes at ~10 ms. **b** The time histories of FBD and DBD afterglow surface charge. The surface charge decay follows a similar trend as the emission intensity, indicating that it is the large surface charge that accumulates on the surface of PZT helps sustain the plasma afterglow. **c** The time-resolved measurements of electrode temperature in FBD and DBD exclude the possible influence of temperature evolution on the sustained FBD since the temperature remains unchanged for a much more extended period. Also, the higher temperature of the FBD electrode than that of the DBD electrode, makes FBD plasma a suitable candidate for the applications (e.g., plasma catalysis) where heat is needed for high efficiency.

estimated to be on the order of ~$10^{-10}$ C cm$^{-2}$, which is negligible compared to the total surface charge of ~$1.5 \times 10^{-7}$ C cm$^{-2}$ of the PZT barrier electrodes shown in Fig. 2. This again confirms the dominant role of PZT's ferroelectricity in inducing large surface charge in FBD.

The ferroelectrics-induced surface charge not only affects the electric field, but also affects the discharge in the time domain. Figure 3d shows the time histories of plasma emission intensity for both FBD and DBD. As shown, the FBD electrode surface charge can better sustain the plasma when the external electric field polarity changes. In each polarity cycle, the time duration of the FBD plasma is apparently more prolonged than that of the DBD plasma. This indicates that the hysteresis nature of surface charge partially retains the streamers between electrodes, and temporally extends the afterglow. By contrast, the DBD with no hysteretic surface charge only lasts for a shorter time duration. In addition, it is noteworthy that Fig. 3d provides further proof of the enhanced plasma brought about by the FBD design. The FBD plasma displays a much stronger emission intensity than the DBD plasma, indicating a much higher ionization rate with higher concentrations of electrons, ions, and excited species (e.g., electronically and/or vibrationally).

**Afterglow enhancement due to ferroelectric surface charge**
Figure 4a shows the time history of emission intensities of plasma discharge and afterglow. In the afterglow of FBD, the plasma emission persists, decays gradually, and lasts for ~680 milliseconds (ms). The instantaneous images of the FBD plasma afterglow, as shown in Fig. 4a, at 0, 400 and 700 ms show such a dynamic decay process: the volume of the FBD plasma shrinks gradually and its overall intensity drops quasi-linearly in afterglow. As a comparison, the DBD afterglow quickly decays and extinguishes at ~10 ms (though still longer than the lifetimes of excited species emitting photons because of the capacitor discharge in the AC power supply). This FBD afterglow persistence is reflected by the temporal response of surface charge. Figure 4b shows the temporal profile of the remaining surface charge on the FBD electrode during afterglow. As shown, the large amount of surface

charge on the FBD electrode decays gradually in the afterglow and lasts for ~720 ms, which is close to the decay time (~680 ms) of the FBD plasma. This indicates that the remaining charge on the PZT surface significantly extends the afterglow time by 68 times. By contrast, the DBD electrode only stores minimal surface charge, quickly diffusing and extinguishing the afterglow. In parallel, we also explored FBD afterglow at other pressures including 40, 70 and 100 torr, and the results are shown in Supplementary Fig. 5 in the supplementary information. We observed the FBD afterglow enhancements similar to that at 10 torr. As seen, the afterglow time is ~776 ms at 40 torr, ~553 ms at 70 torr, and 420 ms at 100 torr. Therefore, as the pressure decreases, the afterglow time first increases, then peaks at 40 torr, and decreases afterward, which scales with the surface charge variation at different pressures shown in Fig. 2. This is because, at lower pressures, the recombination and quenching rates of surface charged species (i.e., electrons) and gas-phase excited species (responsible for afterglow) become lower, while their number densities also become lower. The former leads to more surface charges and a longer afterglow time and the latter leads to less surface charges and a shorter afterglow time, forming a tradeoff on the dependence of surface charge and afterglow time on gas pressure. The afterglow time scales with surface charge at varying pressures, because a larger electrode surface charge needs a longer afterglow time to dissipate via recombination with ions into neutral molecules/atoms.

Then, we investigated the effect of the dielectric constant of ferroelectric materials on the afterglow time. This was accomplished by measuring the afterglow on another ferroelectric barrier electrode, i.e., the tetragonal phase BaTiO$_3$ barrier electrode. The tetragonal phase BaTiO$_3$ is also a common ferroelectric material with a higher dielectric constant of 2500 than 1400 of PZT. For the afterglow measurements on the BaTiO$_3$ FBD, all the experimental conditions (e.g., flow rate and 10 torr pressure) or parameters (e.g., dimensions) remain the same as those for either PZT FBD or alumina DBD, except the electrode barrier material itself. The results were summarized in Supplementary Fig. 6. As seen, with a dielectric

constant of 2500, the FBD with BaTiO₃ barrier electrodes had an afterglow time of ~1330 ms (longer than 680 ms of PZT), and a surface charge of ~0.16 μC cm⁻² (larger than PZT's ~0.11 μC cm⁻²) along with a charge decay time of 1350 ms (longer than PZT's 720 ms). This is because, with the same voltage applied onto the electrodes, the charge on the electrode surface is proportional to the ferroelectric electrode capacitance which is further proportional to the dielectric constant of the electrode. Therefore, a larger dielectric constant of the ferroelectric electrode leads to a larger surface charge which needs a longer afterglow time to dissipate via recombination.

In addition, we confirmed the long afterglow time is an inherent feature of FBD not paraelectric electrode which though also has a dielectric constant much larger than alumina. Again, this paraelectric material took STO (e.g., with a dielectric constant of 300) as an example. Again, the experimental conditions (e.g., flow rate and 100 torr pressure) remain the same as those used on PZT FBD, except the electrode barrier material. The results are shown in Supplementary Fig. 4b. As seen, the plasma discharge with the STO barrier electrodes exhibits a short afterglow time of ~20 ms and a small surface charge of 0.03 μC cm⁻². This is because STO is a paraelectric material which features negligible surface charge when the applied voltage is zero, which is similar to alumina but significantly different from PZT with the spontaneous electric polarization and hysteresis effect. Again, this illustrates that the long afterglow of FBD is not only attributed to the large dielectric constant but also the hysteresis nature formed by the ferroelectric spontaneous polarization.

Then, we also confirm that the surface charge is indeed caused by PZT ferroelectricity instead of piezoelectricity. Figure 4c shows the time-dependent measurements of surface temperatures of the FBD and DBD electrodes via IR thermal imaging (FLIR One Gen 3). As seen, either the FBD or DBD electrode maintains a stable temperature during the entire afterglow decay process, indicating that the electrode has negligible thermal shrinkage. This excludes the possibility of piezoelectrics-induced (including pyroelectrics-induced) surface charge, and leaves ferroelectrics-induced surface charge only. This is further confirmed by measuring the surface charge of a piezoelectric barrier electrode. This work used a single crystal SiO₂ barrier electrode as an example, and the results are shown in Supplementary Fig. 7. Again, except the barrier material itself, all the other experimental conditions (e.g., flow rate and 10 torr pressure) or dimensions for the SiO₂ barrier electrode remain the same as those for PZT FBD. As seen, the plasma discharge with the piezoelectric single crystal SiO₂ barrier electrodes do not exhibit apparent surface charge at each tested pressure, which is considerably different from the large surface charge

of PZT FBD. Instead, the surface charge at each pressure remains at a very low level that is similar to that of the alumina DBD. These comparisons again confirm that the surface charge was indeed caused by PZT ferroelectricity instead of piezoelectricity.

Also, Fig. 4c shows that the temperature of the FBD electrode is apparently higher than that of the DBD electrode, which indicates that the PZT electrode generates more heat than the alumina electrode. This is because the FBD induces a higher current through the electrode, leading to more vigorous Joule heating, compared to the DBD electrode. This feature is expected to push the FBD plasma towards the application where heat is needed for catalyst activation and high-efficiency manufacturing, e.g., plasma catalysis and material synthesis. Here it is notable that the surface charge of the FBD (or DBD) electrode in afterglow maintains a cycling fashion with the same frequency (i.e., 20 kHz) as that before shutdown, as shown in Fig. 4b. We attribute this phenomenon to the resonance between the plasma circuit (containing electrodes, near-electrode sheaths and plasma itself) and the inner circuit of the AC power supply[39,40]. With this resonance, the decayed charge still runs back and forth between the plasma cell and power supply at a frequency determined by the inner circuit of the power supply (i.e., 20 kHz as demonstrated in this work), even in the afterglow.

## Effect of the ferroelectrics on plasma instability

To study the effect of the ferroelectric electrode on plasma instability, we used a gated camera to take instantaneous images of the N₂ FBD and DBD under 10 torr, 40 torr, 70 torr and 100 torr with the same applied voltage, as shown in Fig. 5. Two interesting observations can be made. Firstly, the FBDs are much more unstable with streamers non-uniformly distributed in space, while the DBDs are relatively stable and the streamers therein are more uniformly distributed, similar to the observation in the reference work[24]. The streamers in the FBD plasma are randomly distributed in space at 10 torr only, resulting in a volume-like discharge, and then become increasingly more concentrated in space as the pressure increases. By contrast, the streamers in DBD plasma stay randomly distributed at all pressures. This comparison indicates that FBD provides rich dynamics for streamer propagation, leading to a more energetic plasma. The unstable trend of FBD can be due to two possible reasons. The first possible reason is that the strong surface charge induced by PZT triggers plasma thermal instability[41,42]. The significantly increased charge enhances the local electric field peak, the electron temperature and the number density of electrons therein, leading to more Joule heating and a higher reduced electric field $E/N$ which gives positive feedback to the growth of electron

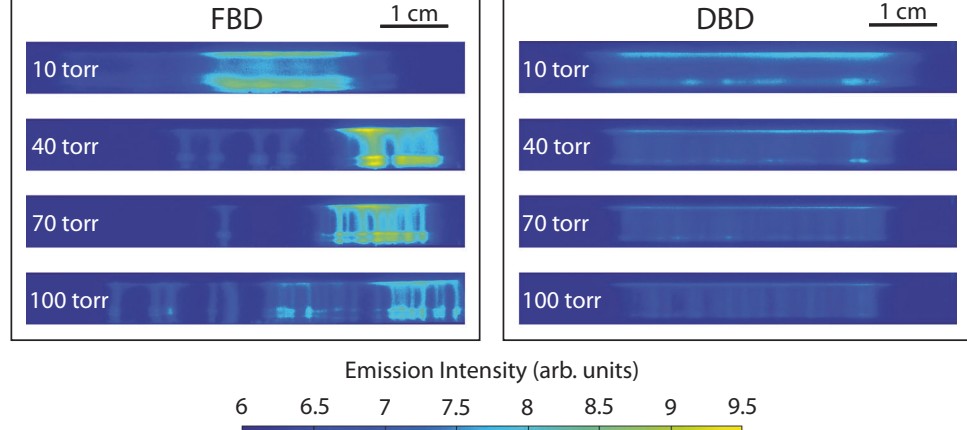

**Fig. 5 | Surface charge induces plasma instability.** The ferroelectric barrier discharge (FBD) is much more unstable with increasing pressures, where the streamers are non-uniformly distributed in space, while the dielectric barrier discharge (DBD) is relatively stable, and the streamers therein are more uniformly distributed. Besides, the FBD plasma is much more intense, indicated by the higher emission intensity, than the DBD plasma.

number density, i.e., the thermal instability. At the same time, with a much higher current (shown in Fig. 3b), the FBD electrode itself generates a significant amount of heat (as indicated in Fig. 4c), which further increases the **E/N** and enhances the instability. The second possible reason is that the surface charge of the FBD electrode may directly induce instability. As the pressure increases, the secondary emission on the surface of the FBD electrode becomes more intense, and any small local disturbance would cause the secondary emission to grow locally and finally result in non-uniformly distributed and unstable streamers. In addition to the non-uniform distribution of streamers, the FBD plasma is also much more intense than the DBD plasma. More intuitively, the FBD plasma is brighter (i.e., stronger optical emission) than the DBD plasma. This is because the surface charge leads to a high current through the plasma, enhancing electron density and temperature in the plasma. Then, the electron impact collisions become more intense, which leads to stronger electronic excitation and generates more excited species (i.e., $N_2(B)$, $N_2(C)$) with more potent optical emissions.

In conclusion, this work reports a plasma discharge with a significantly enhanced electric field and afterglow through a ferroelectric $PZT(Pb(Zr_xTi_{1-x})O_3)$ barrier electrode design. Due to its asymmetric crystal structure, the ferroelectric PZT characterizes spontaneous electric polarization and induces a large surface charge of $10^{-1}$ μC cm$^{-2}$, two orders of magnitude higher than that with the dielectric alumina barrier electrode. The fast polarization of ferroelectric materials enhances the electric field during breakdown in streamer discharge by a factor of two, compared to the DBD, as experimentally proved via in-situ time-resolved EFISH electric field measurements. Also, due to the ferroelectric surface charge, the FBD plasma becomes more temporally sustainable when the external electric field polarity is alternated, and the afterglow time (lasting for 680 ms) is considerably extended by 68 times, compared to the conventional DBD plasma. Lastly, the large ferroelectrics-induced surface charge leads to a high current and significantly causes plasma instability. As the pressure increases, the FBD streamers become much more unstable and non-uniformly distributed in space, while the DBD streamers remain relatively stable and quasi-uniformly distributed. Overall, the FBD provides a sustainable and energy-efficient candidate for various applications including plasma catalysis and material synthesis. In the future, we envision that FBD is highly tailorable. By tailoring properties (e.g., defects) and dimensions of ferroelectrics, FBD can be readily designed towards high-selectivity catalysis and synthesis.

## Methods

### Plasma cell
The plasma was generated in a plate-to-plate fashion. Both the ferroelectric lead zirconate titanate ($Pb(Zr_xTi_{1-x})O_3$, PZT) (Steiner &Martins, Inc.) and dielectric $Al_2O_3$ (McMaster-Carr) are processed to the same dimensions shown in Table 1. These two electrodes are separated by a 5-mm distance and sealed in a 110 mm-long, 24 mm × 11 mm rectangular quartz channel with a wall thickness of 2 mm. In all the experiments, both FBD and DBD are operated with an AC power supply (PVM500/DIDRIVE10, Amazing1) running at a frequency of 20 kHz and a peak-to-peak voltage of 3 kV. $N_2$ gas (Airgas USA, LLC) flows into the quartz cell with a residence time of 0.05 s between electrodes at low pressure ranging between 10 and 100 torr. Here $N_2$ is used for demonstration purposes, and the concept of FBD is also applicable to other gases (e.g., argon).

### Electrical measurements of the discharge and barrier layer
A reference capacitor $C_{ref} = 10$ nF (Digi-Key, 399-12395-ND) is placed in series of the plasma cell. The applied voltage $U_{ext}$ and the reference capacitor voltage drop $U_{ref}$ are monitored using two high voltage probes (Tektronix, P6015A) with a digital oscilloscope (Tektronix, TDS 2012B). The surface charge on the ferroelectric electrode is equal to

**Table 1 | Parameters of the materials in the plasma configuration**

| Part | Materials | Dielectric constant | Dimensions[a] |
|---|---|---|---|
| Barrier layer | PZT | $\varepsilon = 1400$ | $S = 4.5$ cm × 1.7 cm $d = 0.12$ cm |
| | $Al_2O_3$ | $\varepsilon = 9$ | $S = 4.5$ cm × 1.7 cm $d = 0.12$ cm |
| | STO | $\varepsilon = 300$ | $S = 4.5$ cm × 1.7 cm $d = 0.12$ cm |
| | BTO | $\varepsilon = 2500$ | $S = 4.5$ cm × 1.7 cm $d = 0.12$ cm |
| | $SiO_2$ | $\varepsilon = 4.5$ | $S = 4.5$ cm × 1.7 cm $d = 0.12$ cm |
| Metal electrodes | Stainless steel | – | $S = 3$ cm × 0.7 cm |
| Gas gap | $N_2$ | $\varepsilon = 1.00058$ | $d = 0.5$ cm |

[a]S: area facing the discharge region; d: height perpendicular to electrode.

that on the reference capacitor which was obtained by its capacitance times its voltage drop. The total current through the plasma and the unavoidable parallel parasitic components is monitored with the current monitor (Pearson 6585). The equivalent circuit of the system is shown in Fig. 6a.

### Instantaneous images of plasma instability
Single-shot broadband plasma emission images are taken using a gated ICCD camera (Princeton Instruments PIMAX 1300, UNIGEN II intensifier) with a camera lens (Nikon Nikkor 50 mm f/1.4) using a 50 μs gate. The camera is synchronized with the AC power supply by a delay/pulse generator (SRS DG645). Both the FBD and DBD plasmas are generated in a continuous mode with an applied voltage of 6 kV (peak to peak).

### Electric field-induced second harmonic generation
In the EFISH technique, the electric field measurements are accomplished based on the quadratic relation between the second harmonic signal of the excitation laser and the external electric field strength. The experimental schematic for EFISH is shown in Fig. 6b. The 300 μJ laser pulses generated from a femtosecond Ti:sapphire regenerative amplifier (Coherent Astrella) were focused by a plano-convex lens to the center of the plasma between two electrodes. The regenerative amplifier produces 80 fs pulses with a central wavelength of 800 nm (12 nm bandwidth) at 1 kHz repetition rate. To avoid upstream second harmonic generation, a long pass filter was applied before the plasma cell. The second harmonic signal generated from plasma was separated from the initial laser beam by a dispersive prism as well as a dichroic mirror (that reflects 400 nm and transmits 800 nm light). An ICCD camera (Princeton Instruments PIMAX-4 1024i) combined with a 400 nm bandpass filter (FWHM = 10 nm) was used to collect the clean EFISH signal. At each time delay, 200 frames were captured and 10 exposures per frame for average. The ICCD camera gate was 75 ns. To capture time-resolved signals, a delay generator (SRS DG645) was used to delay the power supply from the laser pulse, and also to synchronize the laser pulse with the camera. The pure plasma emission was recorded by blocking the upstream laser for each time step (2 μs) and subtracted from the EFISH signal. The EFISH calibration was performed by measuring known sub-breakdown electric fields using an AC power supply in the same conditions (e.g., gas and reactor) as studied. Note that the EFISH measurements in this work are spatially averaged measurements, considering the plasma volume is not uniform with thin streamers randomly distributed between electrodes.

### Afterglow measurements
After shutting off the power supply, the plasma afterglow (under 10 torr pressure) emission intensity is captured by a high-speed camera (Photron FASTCAM SA-X) with a rate of 1000 fps and a shutter speed of

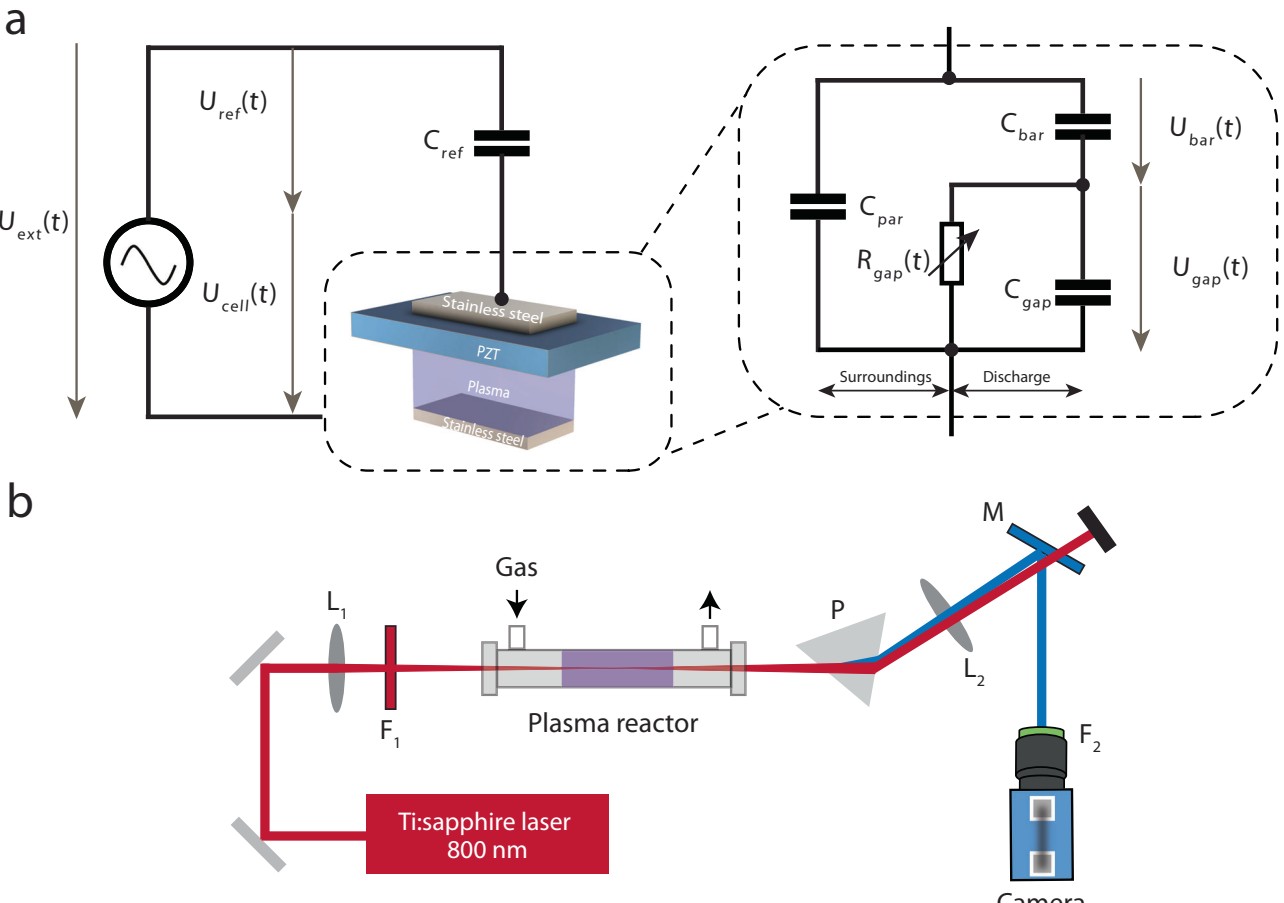

**Fig. 6 | Experiments schematics. a** Equivalent circuit of the electrical measurement. **b** Electric field-induced second harmonic generation (i.e., EFISH) for electric field measurement. $L_1$: plano-convex lens with a focal length of 400 mm; $F_1$: long-pass filter with a cut-on wavelength of 650 nm; P: dispersive prism; $L_2$: collimating lens with a focal length of 750 mm; M: dichroic mirror that transmits 800 nm and reflects 400 nm; $F_2$: bandpass filter with a bandwidth of 10 nm centered at 400 nm.

1/2000 s. The electrode temperature is evaluated by the thermal camera (FLIR One Gen 3).

## Reporting summary
Further information on research design is available in the Nature Portfolio Reporting Summary linked to this article.

## Data availability
The manuscript and supplementary information file present all relevant data that support this study's findings. Source data are provided with this paper.

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

## Acknowledgements

This work was supported by the DOE BES EERC grant DE-AC0209CH11466, DOE grant DE-SC0020233 of Plasma Science Center, DOE grant DE-SC0021135, and NSF grant EFMA 2029425. This publication was supported by the Princeton University Library Open Access Fund.

## Author contributions

Y.X., N.L. and Y.L. designed and performed the experiments. Y.X. and N.L. analyzed the experimental data and prepared the paper draft. X.M., H.Z. and M.N.S. provided theoretical input on data interpretation. Z. C. provided help in performing experiments. Y.J. and N.L. supervised the research. All authors discussed the results and the paper.

## Competing interests

The authors declare no competing interests.
