## [Peer Review File · Nature Communications]

Enhancements of Electric Field and Afterglow of Non-equilibrium Plasma by $\text{Pb}(\text{ZrxTi}(1-x))\text{O}_3$ Ferroelectric ElectrodeREVIEWER COMMENTS

Reviewer #1 (Remarks to the Author):

This manuscript reported a plasma discharge with a significantly enhanced surface charge, electric field during breakdown and afterglow through a ferroelectric PZT barrier electrode design. The authors measured the electric field via measures the electric field and proved that compared to the DBD, the fast polarization of ferroelectric materials enhances the electric field during breakdown in streamer discharge by a factor of two. Afterglow measurements indicated that the FBD plasma becomes more temporally sustainable when the external electric field polarity is alternated, and the afterglow time (lasting for 680 ms) is considerably extended by 68 times, compared to the conventional DBD plasma. The above results are interesting and not reported in literatures although many papers on applying ferroelectric materials to improve plasma manufacturing yield were published. Before the manuscript can be accepted for publication in Nature Communication, some small issues should be addressed.

1. Was the PZT prepoled or not?
2. Why only the afterglow under 10 torr pressure was measured? Under 40-100 torr pressure, did the afterglow time change? The peak surface charge under 40 torr pressure is the largest, how about the corresponding afterglow time? Is the afterglow time related to surface charge?
3. Why the afterglow under 10 torr pressure was measured, but electric field measurements were carried out at 100 torr?
4. The PZT with dielectric constant of 1400 was used in this work. Does the dielectric constant of ferroelectric materials affect the afterglow time?

Reviewer #2 (Remarks to the Author):

This is a very nice work for the PZT FE material application in plasma field. It can be published in NC and no modifications are required.

Reviewer #3 (Remarks to the Author):

Manuscript Number: NCOMMS-23-59598

Title: Enhancements of Electric Field and Afterglow by Ferroelectric Electrode

Incorporating ferroelectric materials into plasma manufacturing is a great idea to enhance the electric field and afterglow of non-equilibrium plasma. A significant enhancement of surface charge, electric field during breakdown, and afterglow by ferroelectric PZT barrier discharge. Due to PZT asymmetric crystal structure, the ferroelectric PZT characterizes spontaneous electric polarization and induces a large surface charge, two orders of magnitude higher than that with the dielectric alumina barrier electrode. The fast polarization of ferroelectric materials enhances the electric field during breakdown in streamer discharge by a factor of two, compared to the DBD. Due to the ferroelectric surface charge, the FBD plasma becomes more temporally sustainable when the external electric field polarity is alternated, and the afterglow time is considerably extended by 68 times, compared to the conventional DBD plasma. There results are very interesting to understand the ferroelectrics-induced surface charge augments electric field and instabilities in plasmas. The paper could be accepted for publication in NC with minor revision. Some issues must be

addressed before accepted for publication.

1. Title of the paper could be revised "Enhancements of Electric Field and Afterglow of non-equilibrium plasma by $\text{Pb}(\text{Zr}_x\text{Ti}_{1-x})\text{O}_3$ Ferroelectric Electrode"

2. Secondary electron emission is very different between PZT and Al_2O_3 , what does play the role of secondary electron emission on the energetic particles? On the other hand, both of them have different work function, the concentration of induced electrons is different, which should be discussed.

3. How to define and detect the seed electrons?

4. It is very difficult to confirm the influence of dielectric permittivity on the electric field and afterglow of non-equilibrium plasma. In this work, the dielectric permittivity of ferroelectric PZT is more than 1000 while that of dielectric Al_2O_3 is about 10, it is better to make a TiO_2 (paraelectric material, dielectric permittivity is ~ 100) or SrTiO_3 (paraelectric material, dielectric permittivity is ~ 300) as comparison.

5. "The ferroelectrics-induced surface charge not only affects the electric field, but also affects the discharge in the time domain." The switching current of ferroelectric domains induced by surface charge can not be seen in this paper.

6. The surface charge could be caused by PZT ferroelectricity instead of piezoelectricity. A comparison should be given. Such as SiO_2 single crystal (piezoelectric material rather than ferroelectric material) or $\text{BaTi}_{0.7}\text{Zr}_{0.3}\text{O}_3$ (paraelectric material rather than ferroelectric material at room temperature).

7. If temperature increases or decreases during creating plasma, the pyroelectricity of PZT should be considered.

Dear Editor,

We thank both you and the reviewers for the help with our manuscript. The manuscript was reviewed by three reviewers. All three reviewers recognized the value of this work. One reviewer recommended direct acceptance, and the other two reviewers recommended acceptance with minor revision. All their comments have been well addressed. The rest of this letter lists our responses point by point, and all the revisions were highlighted in the revised manuscript.

Reviewer 1:

1. Was the PZT prepoled or not?

Yes, the PZT used in this work was pre-poled to achieve satisfactory ferroelectricity in FBD.

The corresponding revision was highlighted in red in paragraph 1 on page 4.

2. Why only the afterglow under 10 torr pressure was measured? Under 40-100 torr pressure, did the afterglow time change? The peak surface charge under 40 torr pressure is the largest, how about the corresponding afterglow time? Is the afterglow time related to surface charge?

We thank the reviewer for the constructive comment. We did not present the afterglow times at 40-100 torr, because we previously thought the afterglow results at those pressures resemble the demonstration result at 10 torr.

After being inspired by this comment, we now recognize the significance of comparing afterglow times with varying pressures. We made up the afterglow measurements at different pressures, and the results are shown below. The afterglow time varies at different pressures, and it is ~776 ms at 40 torr, ~553 ms at 70 torr, and ~420 ms at 100 torr. Therefore, as the pressure decreases, the afterglow time first increases, then peaks at 40 torr, and decreases afterward, which scales with the surface charge variation at different pressures shown in Fig. 2. This is because, at lower pressures, the recombination and quenching rates of surface charged species (i.e., electrons) and gas-phase excited species (responsible for afterglow) become lower, while their number densities also become lower. The former leads to more surface charges and a longer afterglow time, and the latter leads to less surface charges and a shorter afterglow time, forming a tradeoff on the dependence of surface charge and afterglow time on pressure. The afterglow time scales with surface charge at varying pressures, because a larger electrode surface charge needs a longer afterglow time to dissipate via recombination with ions into neutral molecules/atoms.

The corresponding revision was highlighted in red in paragraph 2 on page 10.

Figure R1. The time histories of FBD and DBD afterglow emission intensity and surface charge.

3. Why the afterglow under 10 torr pressure was measured, but electric field measurements were carried out at 100 torr?

We performed EFISH measurements at 100 torr instead of 10 torr, because the EFISH signal at 10 Torr is too low to detect. Theoretically, the EFISH signal is quadratically proportional to the number density of molecules in the plasma. When the pressure decreases by 10 times from 100 Torr to 10 Torr, the EFISH signal becomes 100 times weaker. The detection limit of EFISH at 10 Torr was experimentally tested to be 40 kV/cm (~100 times worse than the detection limit of 0.41 kV/cm at 100 Torr), which cannot be practically feasible to enable the comparison of FBD and DBD like Fig. 3c.

In the meantime, we also measured the afterglow at 100 Torr (shown in Fig. R1) for discussion consistency. The results show that FBD still features a long-lived afterglow compared to DBD, although the FBD afterglow becomes shorter at 100 Torr compared to 10 Torr.

The corresponding revision was highlighted in blue in paragraph 3 on page 8.

4. The PZT with dielectric constant of 1400 was used in this work. Does the dielectric constant of ferroelectric materials affect the afterglow time?

We thank the reviewer for the constructive comment. In this work, we mainly compared the FBD with PZT against DBD with alumina to illustrate the significance of ferroelectricity. Inspired by this comment, we confirmed the influence of the dielectric constant on the afterglow by testing on the

tetragonal phase BaTiO₃ barrier electrode, which is also a common ferroelectric material with a higher dielectric constant of 2500 but a lower curie temperature of 130 °C (which might inhibit its potential application in manufacturing). Except for the electrode barrier material itself, all the other test conditions (e.g., flow rate and 10 torr pressure) or parameters (e.g., dimensions) for the electrode remain the same as those for either PZT FBD or alumina DBD. The results are shown below. The top panel shows the time history of afterglow emission intensity, while the bottom panel shows the time history of surface charge on the BaTiO₃ barrier electrode. As seen, with a higher dielectric constant of 2500 than PZT's 1400, the FBD with BaTiO₃ barrier electrodes had a longer afterglow time of 1330 ms (than PZT's 680 ms), and a larger surface charge of ~0.16 μC/cm² (than PZT's ~0.11 μC/cm²) along with a longer charge decay time of 1350 ms (than PZT's 720 ms). This is because, with the same voltage applied to the electrodes, the charge on the electrode surface is proportional to the ferroelectric electrode capacitance, which is further proportional to the dielectric constant of the electrode. Therefore, a larger dielectric constant of the ferroelectric electrode leads to a larger surface charge which needs a longer afterglow time to quench.

The corresponding revision was highlighted in red in paragraph 3 on page 10 and paragraph 1 on page 11.

Figure R2. Time histories of BaTiO₃ barrier discharge afterglow emission intensity and surface charge.

Reviewer 2:

This reviewer recognized the value of this work and suggested acceptance as it is.

Reviewer 3:

1. Title of the paper could be revised “Enhancements of Electric Field and Afterglow of non-equilibrium plasma by Pb(ZrxTi(1-x))O₃ Ferroelectric Electrode”

We thank the reviewer for the constructive suggestion. We have followed this suggestion as shown in the revised manuscript and Supplementary Information.

The corresponding revision was highlighted in red in the Title on page 1 in both the revised manuscript and Supplementary Information.

2. Secondary electron emission is very different between PZT and Al₂O₃, what does play the role of secondary electron emission on the energetic particles? On the other hand, both of them have different work function, the concentration of induced electrons is different, which should be discussed.

As discussed in ¹, the secondary electron emission in FBD including PZT as electrode barrier material is apparently stronger than that in DBD under the same conditions (e.g., applied voltage and frequency). As a result, FBD contains more free energetic electrons with higher electron temperatures (with higher electric field at breakdown) than DBD, and these electrons collide with neutral molecules, leading to more energetic excited species (electronic and/or vibrational) and more ions in FBD than DBD.

PZT and Al₂O₃ have different work functions which induce different concentrations of electrons. PZT has a work function of ~4.5 eV², smaller than that (i.e., ~4.7 eV³) of Al₂O₃. As a result, the surface of

PZT barrier electrodes in FBD directly emits more electrons into the plasma than that of Al_2O_3 barrier electrodes in DBD, which further induces stronger secondary electron emission and eventually leads to a higher electron concentration in FBD than in DBD.

The corresponding revision was highlighted in red in paragraph 2 on page 5.

3. How to define and detect the seed electrons?

The seed electrons are defined as the initial free electrons before gas breakdown and plasma formation, which are the remaining electrons generated from the last plasma discharge after partially quenching. In repetitive discharges (e.g., AC or pulsed discharge), seed electrons could be accumulated from discharge cycle to cycle. Therefore, the seed electrons of the background gas can accumulate to a certain level (typically on the order of 10^{10} cm^{-3} without FBD Ref. ⁴⁻⁶) and then play a role in the gas breakdown, as we discussed in Fig. 3c.

Generally, if the number density of seed electrons is higher than 10^{12} cm^{-3} , such as that in FBD, they may be detected using multiple methods, including Thomson scattering⁷, Laser Collisional Induced Fluorescence⁸, and Microwave Interferometry⁹, etc. Measurements of seed electrons in FBD and how FBD affects the production of seed electrons using Thompson scattering¹⁰⁻¹¹ are our ongoing work and will merit a separate publication in the future.

The corresponding revision was highlighted in red in paragraph 3 on page 7.

4. It is very difficult to confirm the influence of dielectric permittivity on the electric field and afterglow of non-equilibrium plasma. In this work, the dielectric permittivity of ferroelectric PZT is more than 1000 while that of dielectric Al_2O_3 is about 10, it is better to make a TiO_2 (paraelectric material, dielectric permittivity is ~ 100) or SrTiO_3 (paraelectric material, dielectric permittivity is ~ 300) as comparison.

We thank the reviewer for the constructive comment. We confirmed the influence of dielectric permittivity on the electric field and afterglow by testing on the paraelectric SrTiO_3 (STO, 300 dielectric constant) barrier electrode and the results are shown below. Except the material itself, all the other experimental conditions (e.g., flow rate and pressure) or parameters (e.g., dimensions) for the STO barrier discharge remain the same as those for either PZT FBD or alumina DBD. As seen, the electric field for the STO barrier discharge lies between the electric fields of PZT FBD and alumina DBD, because the STO's dielectric constant 300 between that of PZT and alumina and the surface charge induced by STO is also between those by PZT and alumina. More specifically, due to an intermediate surface charge level, the electric field overshoot of the STO barrier discharge during the first breakdown reaches $\sim 9 \text{ kV/cm}$, which is smaller than that of PZT FBD but larger than alumina DBD. Correspondingly, the concentration of the remaining electrons offered by the first streamer discharge and serving as seed electrons lies between those of PZT FBD and alumina DBD. As a result, the electric field overshoot of the STO barrier discharge during the following gas breakdown cycles also lies between those of PZT FBD and alumina DBD, i.e., decaying faster than the alumina DBD electric field but slower than the PZT FBD electric field.

In parallel, we also tested the afterglow on the STO barrier electrode, and the results are summarized below. As seen, the plasma discharge with the STO barrier electrodes exhibits a short afterglow time of $\sim 20 \text{ ms}$ and a small surface charge of $0.03 \mu\text{C/cm}^2$. This is because STO is a paraelectric material that features negligible surface charge when the applied voltage is zero (similar to alumina but significantly different from PZT with the spontaneous electric polarization and hysteresis effect). This illustrates that the long afterglow of FBD is not only attributed to the large dielectric constant but also the hysteresis nature formed by the ferroelectric spontaneous polarization.

The corresponding revision was highlighted in red in paragraph 2 on page 8 and paragraph 2 on page 11.

Figure R3. The electric field measurements on FBD, DBD and STO barrier discharge.

Figure R4. The time histories of STO barrier discharge afterglow emission intensity and surface charge

5. “The ferroelectrics-induced surface charge not only affects the electric field, but also affects the discharge in the time domain.” The switching current of ferroelectric domains induced by surface charge cannot be seen in this paper.

We thank the reviewer for the comment. As shown in Fig R5, we have zoomed in on other switching currents of ferroelectric domains during the following streamer breakdown (in addition to the first breakdown) shown in Figure 3b. As seen, the switching current profile during the following streamer breakdown is on the same order (i.e., 0.005 A) as the breakdown current of DBD streamers. However, FBD has multiple (i.e., 4-7) current peaks during the breakdown process rather than only one or two peaks in DBD, indicating more streamers formed by large ferroelectrics-induced surface charges. After the first current peak, there are no similar large switching peaks in the following cycles to be seen, because after the first breakdown, FBD has enough charges (from electrons) to form and sustain streamers, so the charges generated from PZT switching are transferred through streamers rather than being detected as a large current peak.

The corresponding revision was highlighted in red in paragraph 2 on page 6 and paragraph 1 on page 7.

Figure R5. The induced current waveforms on FBD and DBD.

6. The surface charge could be caused by PZT ferroelectricity instead of piezoelectricity. A comparison should be given. Such as SiO_2 single crystal (piezoelectric material rather than ferroelectric material) or $\text{BaTi}_{0.7}\text{Zr}_{0.3}\text{O}_3$ (paraelectric material rather than ferroelectric material at room temperature).

We thank the reviewer for the constructive comment. We confirmed the surface charge was indeed caused by PZT ferroelectricity instead of piezoelectricity by further testing on piezoelectric single-crystal SiO₂ barrier electrode as suggested by this reviewer. Except the material itself, all the other experimental conditions (e.g., flow rate and 10 torr pressure) or parameters (e.g., dimensions) for the SiO₂ barrier electrode remain the same as those for either PZT FBD or alumina DBD. The results are summarized below. As seen, the plasma discharge with the piezoelectric single-crystal SiO₂ barrier electrodes do not exhibit apparent surface charge at each tested pressure, which is considerably different from the large surface charge of the PZT FBD. At each pressure, the surface charge is maintained at a very low level that is similar to that of the alumina DBD. These comparisons again confirm that the surface charge was indeed caused by PZT ferroelectricity instead of piezoelectricity.

The corresponding revision was highlighted in red in paragraph 3 on page 11.

Figure R6. Surface charge measurements of FBD, DBD and SiO₂ barrier discharge at different pressures.

7. If temperature increases or decreases during creating plasma, the pyroelectricity of PZT should be considered.

We thank the reviewer for this constructive suggestion. The temperature change on the PZT barrier electrode surface is negligible and within measurement uncertainty (~ 0.1 °C) during gas breakdown and plasma formation. Therefore, the surface charge of PZT caused by pyroelectricity¹² is on the order of $\sim 10^{-10}$ C/cm², which is negligible compared to the total surface charge of $\sim 1.5 \times 10^{-7}$ C/cm² of the PZT barrier electrodes shown in Fig. 2 in the manuscript.

The corresponding revision was highlighted in red in paragraph 3 on page 8.

Sincerely,

Ning Liu, Ph.D.
 Associate Research Scholar
 Department of Mechanical and Aerospace Engineering
 Princeton University, Princeton, NJ 08544, USA
 Email: nl7@princeton.edu

References

1. Navascués, P.; González-Elipe, A. R.; Cotrino, J.; Gómez-Ramírez, A., Large gap atmospheric pressure barrier discharges using ferroelectric materials. *Plasma Sources Science and Technology* **2019**, *28* (7), 075002.
2. Wang, C.; Cao, D.; Zheng, F.; Dong, W.; Fang, L.; Su, X.; Shen, M., Photocathodic behavior of ferroelectric Pb (Zr, Ti) O₃ films decorated with silver nanoparticles. *Chemical Communications* **2013**, *49* (36), 3769-3771.
3. Zahid, M. A.; Khokhar, M. Q.; Park, S.; Hussain, S. Q.; Kim, Y.; Yi, J., Influence of Al₂O₃/IZO double-layer antireflective coating on the front side of rear emitter silicon heterojunction solar cell. *Vacuum* **2022**, *200*, 110967.
4. Wu, S.; Lu, X.; Pan, Y., Effects of seed electrons on the plasma bullet propagation. *Current Applied Physics* **2013**, *13*, S1-S5.
5. Nie, L.; Chang, L.; Xian, Y.; Lu, X., The effect of seed electrons on the repeatability of atmospheric pressure plasma plume propagation: I. Experiment. *Physics of Plasmas* **2016**, *23* (9).
6. Tarvainen, O.; Ropponen, T.; Thuillier, T.; Noland, J.; Toivanen, V.; Kalvas, T.; Koivisto, H., The role of seed electrons on the plasma breakdown and preglow of electron cyclotron resonance ion source. *Review of Scientific Instruments* **2010**, *81* (2).
7. Funaba, H.; Yasuhara, R.; Uehara, H.; Yamada, I.; Sakamoto, R.; Osakabe, M.; Den Hartog, D., Electron temperature and density measurement by Thomson scattering with a high repetition rate laser of 20 kHz on LHD. *Scientific Reports* **2022**, *12* (1), 15112.
8. Barnat, E.; Frederickson, K., Two-dimensional mapping of electron densities and temperatures using laser-collisional induced fluorescence. *Plasma Sources Science and Technology* **2010**, *19* (5), 055015.
9. Andrasch, M.; Ehlbeck, J.; Foest, R.; Weltmann, K., Electron density measurements on an inductively coupled plasma with a one-port microwave interferometer. *Plasma Sources Science and Technology* **2012**, *21* (5), 055032.
10. Chen, T. Y.; Mao, X.; Zhong, H.; Lin, Y.; Liu, N.; Goldberg, B. M.; Ju, Y.; Kolemen, E., Impact of CH₄ addition on the electron properties and electric field dynamics in a Ar nanosecond-pulsed dielectric barrier discharge. *Plasma Sources Science and Technology* **2023**, *31* (12), 125013.
11. Chen, T. Y.; Rousso, A. C.; Wu, S.; Goldberg, B. M.; Van Der Meiden, H.; Ju, Y.; Kolemen, E., Time-resolved characterization of plasma properties in a CH₄/He nanosecond-pulsed dielectric barrier discharge. *Journal of Physics D: Applied Physics* **2019**, *52* (18), 18LT02.
12. Capineri, L.; Masotti, L.; Ferrari, V.; Marioli, D.; Taroni, A.; Mazzoni, M., Comparisons between PZT and PVDF thick films technologies in the design of low-cost pyroelectric sensors. *Review of scientific instruments* **2004**, *75* (11), 4906-4910.

REVIEWERS' COMMENTS

Reviewer #1 (Remarks to the Author):

The authors have revised their manuscript according to the comments of the Reviewers and now the manuscript can be accepted for publication in Nature Communications.

Reviewer #3 (Remarks to the Author):

The authors addressed well the issues, now the paper can be published in NC